# Genetic and Epigenetic Factors in Ulcerative Colitis: A Narrative Literature Review

**DOI:** 10.3390/genes16091085

**Published:** 2025-09-15

**Authors:** Lavinia Caba, Andreea Florea, Petru Cianga, Vasile Drug, Roxana Popescu, Catalina Mihai, Cristian-Gabriel Ciobanu, Vlad Victor Iacob, Laura Florea, Eusebiu Vlad Gorduza

**Affiliations:** 1Department of Medical Genetics, Faculty of Medicine, “Grigore T. Popa” University of Medicine and Pharmacy, 700115 Iasi, Romania; lavinia.caba@umfiasi.ro (L.C.); eusebiu.gorduza@umfiasi.ro (E.V.G.); 2Faculty of Medicine, “Grigore T. Popa” University of Medicine and Pharmacy, 700115 Iasi, Romania; 3Immunology Department, “Grigore T. Popa” University of Medicine and Pharmacy, 700115 Iasi, Romania; petru.cianga@umfiasi.ro; 4Immunology Laboratory, “St. Spiridon” Hospital, 700111 Iasi, Romania; 5Department of Internal Medicine, “Grigore T. Popa” University of Medicine and Pharmacy, 700115 Iași, Romania; 6Emergency Clinical Hospital “Saint Spiridon”, Institute of Gastroenterology and Hepatology, 700111 Iasi, Romania; 7Department of Nephrology-Internal Medicine, Faculty of Medicine, “Grigore T. Popa” University of Medicine and Pharmacy, 16 University Street, 700115 Iasi, Romania

**Keywords:** ulcerative colitis, genetics, epigenetics, GWAS, polymorphisms

## Abstract

Background/Objectives: Ulcerative colitis is a chronic inflammatory bowel disease whose incidence is steadily growing worldwide. The interactions between host genetic susceptibility, gut microbiota and environmental factors determine the onset and relapsing evolution of ulcerative colitis, making it a multifactorial disorder. Methods: A narrative review was conducted to synthesize the available literature on the genes and mechanisms related to ulcerative colitis. Results: The most important independent risk factor is genetics and the candidate genes are associated with inflammation, immune regulation and epithelial permeability. Multiple GWAS have already identified candidate genes and their polymorphisms implicated in ulcerative colitis pathogenesis. Genetic susceptibility is conferred by predisposing variants influencing disease onset and progression, as well as by epigenetic modifications (DNA methylation, microRNAs). Conclusions: This review summarizes the interactions between the functional products encoded by genes linked to ulcerative colitis and immunological factors revealing their common mechanisms.

## 1. Introduction

Inflammatory bowel diseases (IBDs) are complex and heterogeneous conditions that encompass two major and important entities: ulcerative colitis (UC) and Crohn’s disease (CD) [1]. IBD is a multifactorial disorder, characterized by chronic inflammation of different regions of the gastrointestinal tract [2]. According to Kaplan and Windsor, the evolution of IBD can be divided into four distinct epidemiological stages, as follows: emergence stage (the first classified cases of IBD), acceleration in incidence stage (rapid increase in newly diagnosed cases), compounding prevalence stage (the incidence rates are stable, but the prevalence is still growing) and lastly the prevalence equilibrium stage (the prevalence also stabilizes due to stable incidence and mortality rates) [3]. In the last three decades, an increase in the incidence of IBD has been detected worldwide. Even if in some regions an increased incidence was well known, it seems that these diseases are beginning to rise in the newly developed countries, especially in Asia [4,5,6]. A study conducted by GBD 2017 Inflammatory Bowel Disease Collaborators, which analyzed patients’ data from 1990 to 2017, showed an increasing number of affected individuals, from 3,7 million in 1990 to almost double that number (6.8 million) in 2017. The prevalence in the USA was between 252 and 439/100,000 persons [7]. Thus, in the USA, around 1 million people have IBD, while in Europe, it is estimated that around 2.5–3 million people are affected by the disease [5,8]. Zhang and the collaborators evaluated the age-standardized incidence rate for IBD, showing an increased value from 7.98 in 1992 to 8.77 in 2021 [9]. The economic burden of IBD is a major healthcare problem. The annual costs for one patient are very high, exceeding USD 1000 in the USA and reaching up to EUR 10,000 in Europe. These high costs are mostly generated by the treatment, but work absenteeism, disability and premature death generate supplementary costs [10,11,12].

Ulcerative colitis, one of the key phenotypes of IBD, is a common, complex and idiopathic disorder. Over time, a multitude of factors have been determined to contribute to its pathogenesis: genetic susceptibility, host immune system and environmental factors [4,13].

Among these elements, the most important independent risk factor is the genetic one, a fact observed in many families by aggregation studies that have been performed. In individuals with UC, the genetic component presents a rate in the range of 10–15% in monozygotic twins versus 9.4% in dizygotic twins. Moreover, first-degree relatives of IBD patients have a 10-times-higher risk of developing the disease than the general population [13,14,15]. GWAS (Genome-Wide Association Studies) identified more than 240 loci associated with IBD, and many of them (67%) are common to both ulcerative colitis and Crohn disease [15,16,17]. Other risk factors are diet, lifestyle, the components of the microbiome and the use of certain drugs. Smoking has been demonstrated to be a protective factor for UC, as well as appendicectomy before the age of 20 years. Breastfeeding also acts as a protective factor, probably due to its implications in the formation of the gut microbiome at such a young age [4,18,19].

Ulcerative colitis is characterized by its localization restricted to the colon and rectum, with damage only to the mucosa and submucosa layers and a continuous, circumferential and symmetric distribution of lesions on the colon wall. The most common onset is at an age range of 15–30 years [20].

The hallmark symptom in UC is frequent bloody diarrhea. Other symptoms include constipation, abdominal discomfort and the feeling of imminent defecation. Weight loss can occur, as well as extraintestinal manifestations, which are present in 10–30% of cases [20,21]. Extraintestinal occurrences consist of arthritis, uveitis, episcleritis, erythema nodosum, pyoderma gangrenosum and an elevated risk of thromboembolic events, but the most relevant occurrence remains primary sclerosing cholangitis associated with the development of colorectal cancer [1,20,21]. The natural course of IBD includes periods of remission and exacerbations [20]. Complications such as strictures, which can lead to bowel obstructions and even perforations, fistulas and colorectal cancer, may appear [21].

The diagnostic is mainly based on aspects of the clinical presentation and endoscopic and histopathological features, but some laboratory tests also help to exclude other disorders [2,22].

For a correct assessment of the disease, biopsies should be taken from at least two different areas of the lesions and from all affected segments [23,24]. UC might display infiltrates in the lamina propria, micro-abscesses that can fuse and erode the mucosa, causing the formation of pseudopolyps, crypt deformations and crypt abscesses [21,23,25]. Multiple scores are used for the classification of disease severity (Truelove and Witts criteria, Mayo Clinic score, endoscopic Montreal classification system) [2,21,26].

The therapeutic objective is to induce and maintain remission, with minimal side effects. The treatment includes a diversity of medication classes, such as amino salicylates, immunomodulators, corticosteroids, biologic agents, antibiotics and targeted therapies [21]. Due to the optimization of therapeutic management and the availability of multiple classes of medications, colectomy rates have significantly decreased over the past three decades [27]. The surgical treatment for UC is reserved for refractory diseases and for cases that have evolved towards malignancy [25,26].

The aim of our review is to present the most recent knowledge in a field that has accumulated a lot of information concerning the pathophysiology and genetic changes (both at the genome and epigenome levels) in correlation with modern therapies.

## 2. Pathophysiology

There is still a lot to discover about ulcerative colitis pathophysiology, but a few elements have been repetitively evaluated and confirm that a carefully orchestrated interaction of several elements is the key to the onset and progression of ulcerative colitis. The main factors involved are a dysregulated immune response, a disrupted epithelial barrier, gut microbiome changes, genetic susceptibility and the exposome (Figure 1) [28].

### 2.1. Dysregulated Immune Response and Disrupted Epithelial Barrier in Individuals with UC

In healthy individuals, the intestinal epithelium acts as a crucial frontline barrier protecting individuals against invading pathogens. Under normal conditions, there are four compartments of the gut wall that act together to provide equilibrium. The first one is mucus, which lines the colonic epithelium and protects it from toxic substances and dangerous bacteria. The second barrier is formed by intestinal epithelial cells represented by colonocytes, intercalated by goblet cells and enteroendocrine cells. These cells establish a functional barrier by being closely linked to each other via tight junction cells [17,29]. The third compartment is the lamina propria, which includes loose connective tissue and immune cells (macrophages, dendritic cells), while the last compartment is formed solely by mesenchymal cells. In ulcerative colitis, all these compartments are affected in varying degrees, leading to an exaggerated immune response to environmental factors or resident microbiota in genetically predisposed individuals [17].

Immune dysregulation plays a key role in increasing intestinal permeability and compromising barrier function [30]. Ulcerative colitis is marked by abnormalities in both the innate and adaptive immune systems. Antigens stimulate the innate immune response via antigen-presenting cells and T cells, initiating an inflammatory cascade that subsequently engages the adaptive immune system, all these steps finally leading to an increase in pro-inflammatory cytokines secretion, and thus to ongoing inflammation [31]. An imbalance between pro- and anti-inflammatory signals leads to chronic inflammation and damages the intestinal barrier. When pro-inflammatory cytokines dominate, they trigger an excessive immune response, causing increased immune cell activation, increased cell death and a breakdown of the barrier’s integrity [30].

In an undisturbed colon, the active subset of macrophages (CX3CR1 resident macrophages) secretes an anti-inflammatory cytokine (IL-10) that promotes the differentiation and activation of regulatory T cells (Tregs), suppresses pro-inflammatory T helper cell responses and maintains immune equilibrium and the integrity of the epithelial barrier [32,33]. In a study conducted by Hu et al., they observed that chemokine and chemokine-receptor families were positively correlated with M0/M1 macrophages and neutrophils, suggesting that immune cell infiltration and UC hub genes (*CXCL13*, *CXCL10*, *CXCL9*, *CXCL8*, *CCL19*, *CTLA4*, *CD69*, *CD163*, *CCR1*, *PECAM1*, *IL7R*, *TLR8* and *TLR2*) jointly drive disease progression through immune-mediated mechanisms [34]. In UC, there is a shift of the macrophage’s phenotype (CD14+ and CCR2 macrophages) towards a pro-inflammatory action. CCR2 macrophages secret pro-inflammatory cytokines such as TNF-α, IL-1β and IL-6 [32,33]. TNF-α is secreted by macrophages, as well as by other cells (T lymphocytes, B cells, dendritic cells, NK cells, mast cells, endothelial cells and fibroblasts), and its over-expression in ulcerative colitis is correlated with disease activity and severity. It amplifies inflammation by activating other cells, by upregulating some cytokines (Il-1β and IL-6) and by stimulating NF-κB—an indispensable pathway in ulcerative colitis pathogenesis. TNF-α overstimulation also leads to oxidative stress and cellular damage and breaks down epithelial tight junctions [35,36,37]. IL-6 is a key inflammatory mediator that increases epithelial permeability, promotes macrophage infiltration and worsens ulcerative colitis, mainly by modulating the balance between Th17 and Treg cells [31,38,39].

Dendritic cells are antigen-presenting cells, and in ulcerative colitis their function is dysregulated. They also produce pro-inflammatory cytokines (TNF-α, IL-8 and IL-6) and promote Th1 and Th17 differentiation, leading to excessive T-cell activation and chronic mucosal inflammation [33].

Antigen-presenting cells, such as dendritic cells and macrophages, serve as a bridge between innate and adaptive immunity. Despite their different origins, both cells types express pattern-recognition receptors, like Toll-like receptors (TLRs) and NOD proteins. Activation of these receptors stimulates NF-κB and other transcription factors, promoting inflammatory processes [30].

Neutrophils play a central role in UC pathogenesis by initiating, amplifying and sometimes regulating inflammation. They provoke tissue damage by activating the release of reactive oxygen species (which damage cell membranes) and metalloproteinases (which degrade the extracellular matrix and disrupt epithelial junctions). Neutrophils also secrete pro-inflammatory cytokines (TNF-α, IL-1β) and chemokines (IL-8—a potent chemotactic factor for neutrophils, whose concentration has been associated with ulcerative colitis endoscopic and histological activity) [17,33,40].

The innate lymphoid cell population (ILC) is shifted in UC (increased groups 1–2 and decreased protective group 3) contributing to inflammation and barrier dysfunction. ILC1s secrete IFN-γ and TNF-α, promoting inflammation (Th1-like functions), while ILC2s secrete IL-4, IL-5 and IL-13 and are involved in type 2 immune responses (function like Th2 cells). ILC3s are considered protective (function as Th17 cells) due to their abilities to secrete IL-22, which is an important factor for the epithelial barrier. They also secrete IL-17 and TNF-α [30,33,40].

CD4+ T cells are a highly expressed subset in ulcerative colitis and are represented by Th1, Th2, Th17 and Tregs cells [31]. Th1 and Th2 are involved in defense against intracellular pathogens and Th17 in immunity against intracellular pathogens, and thus inflammation in individuals with UC is the result of disrupting the homeostasis of T cells [41]. Th1 cells secrete IFN-γ, TNF-α and IL-2, leading to chronic inflammation by recruiting immune cells and activating macrophages and neutrophils, while Th2 secrete IL-4, IL-5 and IL-13, being involved in epithelial damage. Th17 cells secrete IL-17A, IL-17F, IL-21, and IL-22, generating neutrophil and macrophage recruitment and promoting tissue destruction and inflammation. Treg cells that are downregulated in UC secrete IL-10 and TGF-β and inhibit Th1 and Th2 responses, being essential for suppressing inflammation [31,33]. TGF-β is an inhibitory cytokine that acts as an anti-inflammatory mediator systemically. However, it may exhibit pro-inflammatory effects locally. In IBD, TGF-β, alongside growth factors, contributes to mucosal healing and protects host tissues from luminal insults [35]. In UC, even histologically inactive regions show inflammation and tissue remodeling similar to active lesions, with increased expressions of TGF-β, vimentin and α-SMA [42].

CD8^+^ T cells play a dual role in UC pathogenesis. Conventional cytotoxic CD8^+^ T cells contribute to inflammation and tissue damage by producing TNF-α and other pro-inflammatory cytokines [28].

In UC, B cells contribute to both immune regulation and inflammation. They include effector B cells (secreting IL-2, IL-4, IFN-γ) and regulatory B cells (secreting IL-10). In UC, there is an increase in IgG-secreting plasma cells, which promotes inflammation by enhancing TNF-α and IL-1β production [33].

### 2.2. Signaling Pathways

To date, there are a few signaling pathways highlighted in the development and evolution of ulcerative colitis: JAK/STAT, TLR4/NF-κB, PI3K/AKT, Notch, Wnt/β-catenin and MAPKs. All these pathways can be modulated by targeted therapies for ulcerative colitis [31].

The JAK/STAT pathway (Janus kinase/signal transducer and activator of the transcription) is a key intracellular signaling mechanism that mediates the effects of over 50 cytokines and growth factors, playing a central role in immune regulation and inflammation [31]. Janus kinases (JAK1, JAK2, JAK3) are non-receptor tyrosine kinases and are involved in multiple processes, like cell growth, cell development and cell differentiation. JAK1 is the kinase partner for the receptors of IL-2, IL-10 and type I interferon. JAK2 associates with receptors of growth hormones, leptin, interferons and other interleukins. JAK3 connects with receptors of IL2, IL4, IL7, IL9, IL15 and IL21 [43]. The mechanism steps are simple: cytokines bind to their receptors, leading to the activation of JAKs, which consecutively induce STAT proteins’ phosphorylation, the latter modulating gene expression and initiating cellular processes [43,44]. Macrophages’ polarizations are induced by STAT phosphorylation: M1 phenotype—by STAT1 (following the influence of the IFN-γ—JAK group), M2 phenotype—by STAT6 (after IL4, IL13—JAK activation) or by STAT3 (after I6, IL10—JAK activation) [32]. Another inflammation mechanism is initialized by the release of Th and Tregs cells, after the activation of the JAK/STAT pathway by IL6, IL12, IL23 and IL27 [31].

A central player in the inflammatory cascade of UC is the NF-κB (nuclear factor kappa-light-chain-enhancer of activated B cells) pathway. It is a key transcription factor that regulates genes involved in inflammation, immune response, cell survival and epithelial homeostasis. NF-κB can be either activated via the canonical pathway by cytokines (TNF-α and IL-1β ultimately promote the transcription of pro-inflammatory genes), microbes and stress, or via the non-canonical pathway through specific receptors (leading to disturbances in the immune system) [31,44,45]. NF-κB pathway activation (by the interaction of Toll-like receptor 4 and lipopolysaccharides) also leads to inflammation, promoting macrophage polarization to a M1 phenotype [32].

The Wnt/β-catenin signaling pathway is crucial for mucosal renewal and for barrier functions; alterations in this pathway lead to abnormal homeostasis and the development of ulcerative colitis. Wnt ligand expression is highly segregated along the intestinal tract, and the dysregulation of Wnt signaling can follow two different directions [44,45]: (1) An aberrant activation (hyperactivation) of the Wnt/β-catenin pathway leads to intense epithelial proliferation and consequently to fibrosis and dysplasia, increasing the risk of colorectal cancer development; (2) Impaired signaling, and thus reduced Wnt/β-catenin activity, triggered by either genetic mutations or cytokines influence. Decreased activity of this pathway contributes to chronic inflammations, due to the inability to support intestinal barrier functions [44].

The mitogen-activated protein kinase (MAPK) pathway is a conserved kinase cascade involved in regulating cell proliferation, apoptosis, differentiation, inflammatory responses and transcriptional regulation. The MAPK family includes stress-activated serine/threonine–protein kinases such as extracellular-regulated kinases 1–8 (ERKs), p38-alpha kinase, Nemo-like kinase (NLK) and c-Jun N-terminal kinase (JNK). Oxidative stress and other extracellular stimuli (growth factors, pro-inflammatory cytokines—TNF-α, IL6, IL-1β) activate MAPKs and consequently regulate gene expression and influence cellular responses. JNK and p38 kinases promote apoptosis. MAPKs are also involved in TLR signaling pathways in dendritic cells [43,44,45].

The PI3K/Akt/mTOR (phosphatidylinositide 3-kinase/protein kinase B/mTOR) pathway is a major intracellular signaling cascade involved in cell survival, metabolism, proliferation and immune regulation. It is activated by signals from receptor tyrosine kinases, G-protein-coupled receptors and Toll-like receptors. The cascade follows this route: PI3K activates Akt, which stimulates mTOR—an indispensable element for cell growth and metabolism. This pathway plays multiple roles, and its dysregulation leads to (1) the weakening of the intestinal barrier via the disruption of tight junction proteins, (2) overactivation of immune responses, followed by continuous inflammation and (3) alterations in host–microbiota interactions leading to deficits in epithelial repair mechanisms [32,44]. Akt1 promotes M2 macrophage polarization, playing an anti-inflammatory role, while Akt2 has an opposite effect (pro-inflammatory), enhancing M1 polarization [32].

AMPK (AMP-activated protein kinase) is a master regulator of cellular energy homeostasis, and its activation plays a protective role in UC through multiple interrelated mechanisms: autophagy regulation, pyroptosis inhibition, anti-inflammatory effects, restoring intestinal barrier functions and macrophage polarization. Autophagy is essential for maintaining intestinal homeostasis, immune balance and barrier integrity [46]. Autophagy also reduces mitochondrial damage and reactive oxygen species production, which are triggers for NLRP3 inflammasome activation [47]. Normally, AMPK activates ULK1, a serine/threonine-protein kinase, which acts both as a downstream effector and negative regulator of mTORC1 (mammalian target of rapamycin complex 1) [43,46]. In ulcerative colitis, deficient autophagy leads to barrier dysfunction, endoplasmic reticulum stress and altered microbiota. Pyroptosis is a form of inflammatory cell death that worsens epithelial injury in individuals with UC. The AMPK pathway plays a protective role in individuals with UC by inhibiting the NLRP3 inflammasome and downstreaming other pyroptotic pathways [46]. NLRP3 is a pattern-recognition receptor that forms a multiprotein inflammasome complex in response to pathogen-associated and danger-associated molecular patterns. It triggers caspase-1 activation and induces the secretion of pro-inflammatory cytokines (IL-1β and IL18). Its upregulation in UC leads to excessive inflammation and mucosal damage [45,47]. In addition to the inhibition of NLRP3 inflammasome formation, the AMPK pathway has an anti-inflammatory effect by suppressing other signaling pathways (NF-κB and MAPK), reducing cytokines (IL-1β, IL18, TNF-α, IL6) and releasing and reducing immune dysregulation (via Th17/Treg balance). By NF-κB inhibition, AMPK also promotes M2 polarization over the M1 phenotype [46]. The Notch pathway is a highly conserved cell–cell communication system that regulates critical cellular functions, such as proliferation, differentiation, apoptosis, development and migration. This pathway includes Notch receptors (Notch receptors 1–4, of which Notch 1 is preponderant in the intestine), ligands (delta-like ligands 1, 3, 4 and jagged 1, 2) and transcriptional machinery [48]. The cascade of events that occurs after ligand delta-like 4 binds to the Notch 1 receptor promotes M1 macrophage polarization. Notch 4 is also implicated in regulating other pathways, like the inhibition of STAT1 and increasing STAT3 activation [32]. The Notch signaling pathway plays a key role in repairing the intestinal mucosal barrier, particularly the chemical and physical components. The Proper and tightly controlled regulation of this pathway is crucial for limiting the infiltration of pathogenic antigens into underlying tissues. In individuals with UC, an abnormal increased activity in the Notch pathway, especially regarding the inhibition of intestinal stem cell differentiation into secretory cells, has been reported. These last cells are key producers of MUC2; therefore, in individuals with UC the mucus barrier starts to weaken. The Notch pathway is also involved in cell communication, its activation leading to compromised tight junctions and adhesion junctions, causing antigen leakage [48].

### 2.3. Gut Microbiota

Gut microbiota (GM) are vast, varied, complex, abundant and stable. The main four phyla of the human gut microbiome are *Firmicutes*, *Bacteroidetes*, *Proteobacteria* and *Actinobacteria* [41]. Intestinal bacteria can be classified into three categories: aerobic, facultative anaerobic and anaerobic. GM play the following beneficial roles in the body’s functions: the development of the immune system, host defense and the production of energy and nutrients [49]. “Microflora hypothesis” states that the normal development of the immune system and homeostasis require commensal microbiota, which plays a role in protecting the host’s immune system from dysregulation [50].

The dysbiosis found in IBD is defined as a quantitative and qualitative microbial imbalance. Dysbiosis can lead to an aberrant immune response [51]. In individuals with UC, studies showed a reduction in microbiota’s diversity and unstable microbiota [51,52]. There are studies that also indicate a decrease in dominant anaerobes and a reduction in the concentration of organic acids in patients with UC. The dominant anaerobes (*Bacteroides* and commensal *Clostridium*) are important for maintaining intestinal homeostasis [51]. The altered gut microbiome in UC is characterized by an increased abundance of *Bifidobacteriaceae* and *Acidaminococcaceae*, while *Propionibacteriaceae* and *Nectriaceae* are notably decreased [53]. Some studies declare no specific bacteria with roles in the development of UC [41], while others determined some species, such as *Bacteroides uniformis* and *Bacteroides bifidum*, to be specifically associated with the disease [53]. The intestinal barrier function is disturbed, which leads to increased permeability at its level, and therefore alterations in innate and acquired immune responses [41,54].

The exposome encompasses the full range of environmental factors experienced by an individual from conception onwards [55]. One of the earliest factors that has a protective impact on ulcerative colitis is breastfeeding. It influences an infant’s microbiota by shaping bacterial diversity and supporting the development of mucosal immunity. Moreover, breast milk contains components that enhance an infant’s immune system, leading to a reduced risk of infections with microorganisms—another independently studied element of exposome [56]. Another protective factor is tobacco-smoking. Numerous studies show that its cessation induces ulcerative colitis flare up [57,58]. Diet, and the modern composition of food, which includes a lot of additives and pollutants, has been implicated in UC flares by disturbing the gut microbiome as well as the gut intestinal barrier [55,59]. The Western diet has an important impact on microbiota diversity in individuals with IBD, leading to barrier dysfunction by promoting the proliferation of certain microorganisms like *Bacteroides thetaiotaomicron*, *Akkermansia muciniphila*, *Escherichia coli* and pathogenic species of *Proteobacteria* [59]. Other environmental risk factors are psychosocial stressors and the use of certain medications, such as non-steroidal anti-inflammatory drugs, oral contraceptive and some antibiotics, which have the same effects on UC evolution as diet [55,59]. Microbial infections also contribute to an elevated risk of UC development by inducing inflammation through several mechanisms, such as the disruption of the intestinal mucosal barrier, inducing dysbiosis and consequently impairing immune regulation [53].

## 3. Genetics and Epigenetics in Individuals with UC

### 3.1. Monogenic UC

Monogenic IBD results from rare, deleterious mutations in genes implicated in the pathogenesis of inflammatory bowel disease (both ulcerative colitis and Crohn’s disease). It typically manifests during early childhood and can be categorized into two groups: infantile IBD (onset within the first 2 years of life) and very early-onset IBD (disease onset before the age of 6 years). The clinical phenotype is generally more severe than in polygenic forms and is frequently associated with recurrent infections [60]. Monogenic defects disrupt intestinal immune homeostasis through impaired epithelial barrier function, defective bacterial clearance by phagocytes, excessive inflammation and abnormal T- and B-cell regulation [61]. The mode of inheritance varies—most genes are autosomal recessive, but there are several X-linked or autosomal-dominant [62]. The main gene implicated in monogenic IBD is the *IL10* gene and *IL10R* gene—autosomal recessive inheritance. Hyperactivation of the immune response may arise from defective inhibitory pathways, particularly impaired IL-10 signaling or regulatory T-cell dysfunction. The crucial role of IL-10 within the colonic mucosa is evidenced by infants carrying mutations in *IL10* (interleukin 10), *IL10RA* (interleukin 10 receptor subunit alpha) or *IL10RB* (interleukin 10 receptor subunit beta), who develop severe colitis within the first weeks of life. Loss-of-function variants in these genes lead to very early-onset IBD with perianal disease and folliculitis, representing a monogenic form of IBD with complete penetrance [61].

Monogenic IBD presents other common alterations in the following genes: *XIAP* (X-linked inhibitor of apoptosis) and *FOXP3* (forkhead box P3) [X-linked inheritance], *FERMT1* (FERM domain containing kindlin 1), *CYBB* (cytochrome b-245 beta chain), *COL7A* (collagen type VII alpha 1 chain) [autosomal recessive inheritance] and *GUCY2C* (guanylate cyclase 2C) [autosomal dominant inheritance]. *FERMT1* and *COL7A* genes contribute to UC pathogenesis through barrier dysfunction, *CYBB* is involved in phagocyte defects and *FOXP3* is involved in impaired regulatory T-cell function. The *XIAP* gene, which inhibits apoptosis, plays a key role in regulating both innate and adaptive immunity [62,63].

### 3.2. GWAS Studies on Ulcerative Colitis

The GWAS Catalog analysis allowed the identification of GWAS that followed the association of polymorphisms and ulcerative colitis. We considered only studies in which the *p*-value showed a statistically significant association between SNP and disease. Appendix A presents the genes in which polymorphisms associated with UC were identified [64]. The roles of the products of these genes play various roles and intervene in innate mucosal defense, autophagy, oxidative stress, epithelial barrier function, immune cell recruitment, T-cell regulation, ER stress and drug bioavailability. Among the best-known and most important genes, we mention the role of the interleukin family (*IL21*, *IL23R*, *IL1R1*, *IL1R2*, etc.) in immunity through the regulation of T helper cells (especially via the Th17 signaling pathway). The *STAT3* gene is responsible for modulating cellular responses to interleukins. Another important family is TNF, whose gene products drive inflammation through the secretion of pro-inflammatory cytokines. The genes *GNA12* and *ERRF11* are among those involved in alterations in the intestinal barrier. *PARK7* and *DAP* genes are implicated in autophagy, with *PARK7* also contributing to oxidative stress alongside the *DLD* gene. *CXCR1* and *CXCR2* genes are involved in immune cell recruitment. Another gene family, *TLRs*—particularly *TLR4*—plays a role in modulating innate immune responses. Finally, *ADCY7* is another gene involved in innate immunity [43,65,66]. GWAS are presented in Appendix A [67,68,69,70,71,72,73,74,75,76,77,78,79,80,81,82,83,84,85,86,87,88,89,90,91].

### 3.3. Rare Variants

Some rare variants are important genomic drivers of IBD, having strong effects. Many such variants have been statistically linked to complex IBD and functionally validated in monogenic IBD. They often overlap with common variants [92]. Rare variants are more likely to be restricted to certain populations. Visschedijk et al. highlighted how a rare variant in the *IL23R* gene (rs76418789) can differ among different populations. They also identified rare variants in the *MUC2* gene that conferred susceptibility to UC in their Dutch cohort, whereas in a German cohort, they did not find an association between this gene and UC. They also confirmed previously reported rare variants in *IL23R* (rs41313262, rs76418789, rs11209026), *CARD9* (rs141992399, rs200735402) and *JAK2* (rs41316003) [93].

Another study by Wu et al. demonstrated an interesting association between rare variants in *ITGB4* (c.C2503G; p.P835A) and *MUC3A* (c.C1019T; p.P340L), which may represent potential causative variants for UC associated with primary sclerosing cholangitis [94]. Another rare variant was reported in two Afro-American patients, which were carriers of p.Tyr185Cys in the *DUOX2* gene [95]. Another study found a rare variant in *NLRP7* (p.S361L) and a missense change in *TRIM31* (p.C48R, rs140451451) to be associated with UC [96]. We only highlighted some of the rare variants found in individuals with ulcerative colitis, since most studies focus on inflammatory bowel disease in general.

### 3.4. HLA Genes

HLA genes are divided into class I (*HLA-A*, *B*, *C*) and class II (*HLA-DR*, *DQ*, *DP*), with the strongest IBD associations seen in class II alleles, particularly *HLA-DRB1* and *HLA-DQB1* [97] (Table 1). The strongest association is with *HLA-DRB1*0103*, which is linked to severe, extensive UC, higher rates of hospitalization, increased systemic corticosteroid use and a greater likelihood of requiring major surgery [97,98]. *HLA-DQA1*05* is positively associated with more extensive colonic inflammation at the time of UC diagnosis in children and adolescents. Carrying the *HLA-DQA1*05* allele doubles the risk of developing immunogenicity to anti-TNF therapy and is associated with an increased likelihood of not having a therapeutic response to ulcerative colitis [98,99].

### 3.5. Epigenetics in UC

Epigenetic changes refer to changes in gene expression, but which do not modify the DNA sequence. The most common epigenetic mechanisms are DNA methylation, histone modification and non-coding RNAs. The main epigenetic modifications in UC are DNA methylation changes and microRNA [66,102].

#### 3.5.1. DNA Methylation

DNA methylation represents the process by which a methyl group is added to a cytosine or adenine at the 5-position of carbon, where the DNA base thymine is located, and in this way, it is transformed into methylcytosine. This process takes place especially in cytosine phosphate guanine (CpG) dinucleotide sequences [65,103].

In individuals with UC, genes involved in homeostasis and defense mechanisms tend to be hypermethylated, whereas those associated with immune responses—such as chemokines and interleukins—often show hypomethylation. Also, anti-inflammatory genes (*IL10*, *SIGLEC5*, *CD86*, *CLMP*, *NLRP3* and *NLRC4*) tend to be hypomethylated in mucosal biopsies from patients with severe UC, compared to the ones with mild UC [104]. Zeng et al. summarized the studies on the role of DNA methylation in individuals with IBD and showed that the methylation status increased in UC patients compared to the healthy controls for the genes *THRAP2*, *FANCC*, *GBGT1*, *WDR8*, *CARD9* and *CDH1*, while the methylation status was low for *ICAM3*, *DOK2*, *TNFSF4* and *VMP1*. In an assessment of disease activity, methylation increased for some genes (*CDH1*, *GDNF*, *SLIT2*, *MDR1*, *FMR1*, *GXYLT2* and *RARB*) or decreased (*FOXA2*, *ROR1*, *NOTCH3*, *CDH17*, *PAD14*, *TNFSF8*, *EPHX1*, *HOXV2* and *FRK*) in patients with active UC versus patients with quiescent UC [105]. In peripheral blood, a hypermethylated status of *CXCL5*, *CXCL14*, *IL4R*, *IL17C* and *GATA3* was observed in UC patients compared with the healthy controls [104].

#### 3.5.2. Histone Modifications

Histone modifications are mediated by three classes of protein: writers, erasers and readers. Writer proteins add chemical groups to histones through processes such as methylation, acetylation and ubiquitination, thereby influencing chromatin structure. Eraser proteins remove these modifications, reversing their effects. Reader proteins recognize specific histone marks and modulate gene expression by recruiting transcription factors and transcription repressors [106]. All these changes lead to gene hyperexpression or inactivation and play a critical role in shaping immune responses in the context of inflammatory disease immunopathogenesis [104,106].

##### Histone Methylation

Histone methylation mainly occurs on lysine and arginine residues. Lysine can be mono-, di- or tri-methylated by methyltransferases (writers), while demethylases (erasers) remove these marks. Histone methylation can either silence genes or promote active transcription. Silencing marks include the trimethylation of H3 at lysines 9, 27 and 36 (H3K9, H3K27, H3K36), and H4 at lysine 20 (H4K20). In particular, H3K9 trimethylation is essential for establishing and maintaining stable heterochromatin [106]. Epigenetic studies indicate that the trimethylation of H3K27 can influence CD4+ T-cell differentiation through gene regulation. JMJD3 (also known as lysine-specific demethylase 6B, KDM6B) presents histone demethylase activity and modulates gene expression by removing methyl groups from H3K27. A JMJD3–H3K27 interaction can target specific gene loci in chromatin, directly altering gene expression through histone demethylation, thereby regulating the differentiation of inflammation-related immune cells and impacting the progression of ulcerative colitis [107].

##### Histone Acetylation

Acetylation occurs only on lysine residues, and its level is controlled by the balance between histone acetyltransferases and histone deacetylases. Lysine acetylation reduces the charge interaction between DNA and histone tails, loosening the chromatin structure [103,106]. In individuals with UC, LYZ, S100P and NPSR1 show an increased expression and H3K27 (histone-3 lysine-27) acetylation, which correlates with higher transcriptional activity in gene promoters and enhancers. In contrast, inflamed UC colons display a reduced expression of lysine acetyltransferase KAT2B. The knockdown of KAT2B decreases its binding and reduces H4K5 acetylation at IL-10 promoter regions, leading to the transcriptional silencing of the anti-inflammatory cytokine IL-10 in inflamed IBD tissues. A further decrease in histone H3 acetylation was observed in the epithelia of UC patients [104].

#### 3.5.3. MicroRNA

In recent years, special interest was paid to the implication of microRNA (miRNA) in the pathogenesis of IBD, suggesting that it could be used equally as a diagnostic marker and as a treatment tool [108]. The regulation of mRNA stability and translation plays an important role in some processes related to IBD, such as immune defense, recovery of the mucosa after injuries and maintaining the integrity of the gut barrier. For the correct control of these processes, an interaction between specific sequences of mRNA and RNA-binding proteins, or non-coding RNA (especially miRNA) is required [109]. MicroRNAs are small, non-coding, single-stranded RNAs that play a crucial role in gene regulation across various diseases. In UC, miRNAs influence disease progression by modulating immune cell function, maintaining the integrity of the intestinal epithelial cell barrier and preserving the balance between the host and gut microbiota [104]. MiRNAs have been associated with the disruption of tight junctions and intestinal mucus barrier, changes in immunological responses and the regulation of apoptosis and autophagy [108].

In a study conducted by Síbia et al., 33 upregulated and 7 downregulated miRNAs were correlated with UC [110]. MiR-155-5p is over-expressed in UC and it is associated with several signaling pathways. It can downregulate the *FOXO3a* gene, which leads to the activation of NFκB. It also interacts with Toll-like receptors IL-1, IL-8, NK cells and Th17 cells, all of these contributing to inflammatory responses in individuals with UC [108,110]. Another upregulated factor is miR-20b, which presents inhibitory effects over the differentiation of Th17 cells and reduces the expression of the *NOD2* gene (which has a well-known contribution to the inflammation process). MiR-223 is also upregulated, plays an important role in Th17 and IL-23 signaling pathways and is associated with disease activity. MiR-874 weakens intestinal barrier function by reducing aquaporin 3, and it is also upregulated in UC [108]. MiR-21-5p has an interesting distribution: it is over-expressed in the colonic tissue and saliva and under-expressed in the blood. Its implications in the permeability of tight junctions, the destruction of the intestinal barrier (by altering the PTEN/PI3K/Akt signaling), the regulation of apoptosis and cell proliferation and the stimulation of the macrophage inflammatory protein 2 (MIP2) and TNF-α make it a key element in the pathogenesis of UC [108,110,111]. MiR-24 is another element that is upregulated in colonic biopsies, especially in patients with active UC, associated with tight junction functions [112]. Another downregulated factor in the blood and upregulated in colonic tissue is miR-146a, which disturbs the function of NK cells (via STAT1 pathway) and inhibits the differentiation and proliferation of Th1 cells (by targeting STAT4 pathway) with the promotion of the inflammation process [108,110]. Another factor implicated in the STATs pathway, more precisely STAT3, is miR-214, thus promoting inflammation and the development of UC [113]. MiR-31-5p also has a dual-pattern distribution: it is over-expressed in the colonic tissue and saliva and under-expressed in the blood. It inhibits the differentiation and function of regulatory T cells [108,110]. Other interesting miRNAs proposed to influence UC evolution are miR-223 (increased), which can reflect disease severity; miR-182-5p, which is implicated in the Wnt/β-catenin signaling pathway, exacerbates UC symptoms and, furthermore, it can be associated with colorectal cancer progression; and miR-195-5p, which may explain steroid resistance in some UC patients [104].

MiRNA shows a promising evolution in the diagnostic and prognosis of patients with IBD. First, it has been shown that its expression can be changed before the onset of IBD symptoms. Secondly, its expression can change during the natural course of the disease, making it easier to differentiate between the different stages of the disease. Furthermore, miRNAs can be extracted from serum and plasma, which makes it a potential non-invasive diagnostic marker [110].

#### 3.5.4. Long Non-Coding RNAs

Long non-coding RNAs (lncRNAs) are transcripts longer than 200 nucleotides and are generally poorly conserved. LncRNAs may be polyadenylated or not, and about 98% undergo splicing, with roughly 25% having at least two alternatively spliced isoforms. Common features of lncRNAs include low expression levels, tissue-specific expression and exonic regions with low interspecies sequence conservation [114]. LncRNAs have diverse functions: they can regulate protein-coding genes through chromatin remodeling, control gene expression at transcriptional or post-transcriptional levels, guide chromatin modification complexes to specific genomic loci and modulate protein activity and stability.

Studies have shown that lncRNAs play important roles in the pathophysiology of IBD [115]. LncRNA BC012900 is upregulated in active UC tissues and is induced by cytokines and pathogens via established IBD-related pathways, including Toll-like and NOD2 receptors [104,115]. The lncRNA CRNDE contributes to intestinal barrier dysfunction by inhibiting miRNA-495 and increasing the expression of the cytokine signaling inhibitor SOCS1. In individuals with UC, miRNA-495 levels are reduced, and miRNA-495 normally protects intestinal epithelial cells from apoptosis via the JAK signaling pathway. LncRNA MEG3 (maternally expressed 3) has anti-inflammatory effects and is implicated in immune dysregulation in individuals with UC. MEG3 appears to inhibit inflammatory cytokine and ROS release by stimulating IL-10, and it can prevent both pyroptosis and apoptosis in individuals with UC. LncRNA ANRIL is downregulated in IBD. The inhibition of ANRIL alleviates LPS-induced damage by negatively regulating miR-323b-5p, which in turn targets TLR4. Therefore, ANRIL may influence UC development by modulating immune and inflammatory responses through the miR-323b-5p/TLR4/MyD88/NF-κB pathway [115]. A study conducted by Wu et al. showed that METTL14 knockdown worsens colonic inflammation in individuals with UC by regulating the lncRNA DHRS4-AS1/miR-206/A3AR axis via the m6A modification of DHRS4-AS1 [116].

## 4. Importance of Genetic Variants in Personalized Therapies

Over time, multiple therapies have been targeted against the pathways of ulcerative colitis development and progression. Recently, advances have been made, and some evidence suggests that genes and their unique polymorphisms may also help us to understand the therapeutic strategies (Table 2). SNPs in *IL23R*, *IL6* and *IL1B* genes and in Toll-like receptor genes (*TLR9*, *TL2*, *TLR4*) have been suggested to modify organism response to anti-TNF drugs, such as infliximab [117,118]. A single gene variant is not enough for the protection from or susceptibility to disease, and each polymorphism has its own contribution. For example, the polymorphism in the *IL23R* gene (rs10889677, rs11209032, AA genotype of rs1004819, rs2201841 GG genotype, CC genotype of rs1495965) has been associated with a positive response to infliximab in patients with moderate to severe UC, while other ones (AA genotype for rs1343151, GG genotype of rs7517847 and rs11465804, rs10489629 CC genotype) modified the response, decreasing responsivity [118]. Salvador-Martin et al. reported an association between *TLR4* rs5030728 and *LY96* rs11465996 with subtherapeutic infliximab trough levels in children with IBD. These variants were not linked to adalimumab levels. Notably, the A allele of *TLR4* rs5030728, previously associated with an improved response to anti-TNF therapy in individuals with UC, was linked to a greater likelihood of achieving therapeutic infliximab levels compared with the GG genotype. Additionally, three variants were associated with altered adalimumab exposure: *TNFRSF1B* rs3397 (lower levels), *TLR2* rs1816702 and *CD14* rs2569190 (higher levels). Instead, homozygous carriers of *TLR2* rs1816702 (T allele) or *CD14* rs2569190 (G allele) showed higher trough levels for both drugs. *TNFRSF1B*, a receptor involved in apoptosis and survival, has been linked to non-response to infliximab and lower mRNA expression in carriers of the rs3397 T allele [119].

Thiopurine drugs, such as azathioprine, are used in refractory disease as well as in steroid-dependent patients, being a useful adjuvant for anti-TNF agents to minimalize anti-drug antibody formation. Polymorphisms in genes like *TPMT* and *NUDT15* have been associated with pharmacokinetics and pharmacodynamics modifications of this drug, leading to toxicity. Moreover, some *NUDT15* polymorphisms are associated with azathioprine-induced leukopenia [120,121,122].

Regarding the HLA variants and therapeutic options, it is worth mentioning that HLA-DRB10103 has been associated with immunogenicity to infliximab in IBD cohorts, while HLA-DQA105 is associated with increased anti-drug antibody formation and therapy failure [98].

Emerging evidence suggests that lncRNAs play a pivotal role in UC pathogenesis, opening new avenues for therapy. The importance of lncRNA DHRS4-AS1 via the miR-206/A3AR axis identified *METTL14* as a potential therapeutic target [116]. Additionally, the suppression of ANRIL may inhibit UC progression via the miR-323b-5p/TLR4/MyD88/NF-κB pathway, highlighting this axis as another promising strategy for UC treatment [115].

These are just a few examples of how gene polymorphisms can significantly influence treatment response, toxicity and the personalization of UC therapy. The probability of discovering new variants with therapy implications is endless, but more studies are required in this field.

## 5. Conclusions

Ulcerative colitis is a multifactorial disease whose pathogenesis involves host factors (genetic factors, immunity, gut microbiota) and environmental factors. Any imbalance of these elements produces irregularities that result in the appearance of inflammatory bowel disease. GWAS have made it possible to dissect the relation between the genes implicated in ulcerative colitis and the main mechanisms. It is important to know the genetic factors involved in the pathogenesis of UC because this is the premise for the identification of biomarkers useful for early diagnosis and personalized treatment.

## Figures and Tables

**Figure 1 genes-16-01085-f001:**
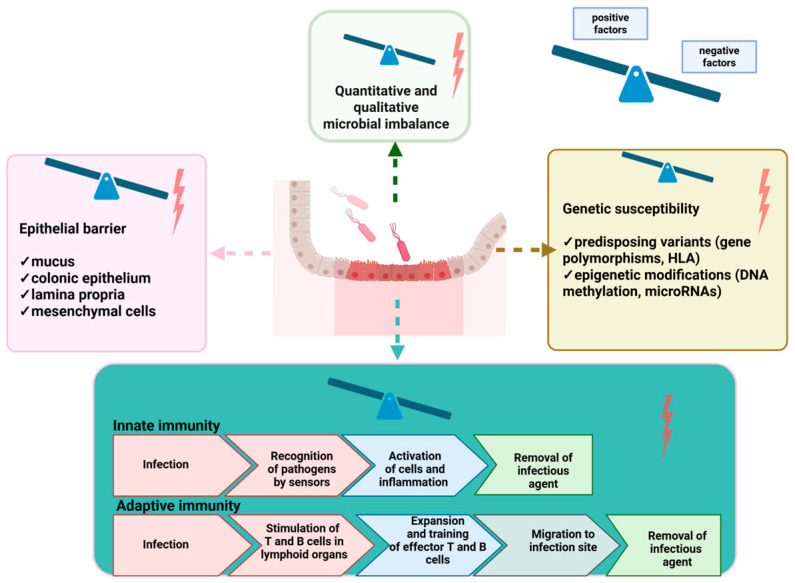
Pathophysiology in UC (Created in BioRender. Caba, L. (2025) https://BioRender.com/c7fmror accessed on 29 July 2025).

**Table 1 genes-16-01085-t001:** HLA variants and risk allele associated with UC identified by GWAS [64].

Variant and Risk Allele (rs)	*p*-Value	OR	CI	Gene	References
rs9271209-?	2 × 10^−85^	-	-	*HLA-DRB1*, *HLA-DQA1*	[79]
rs76901167-?	2 × 10^−94^	-	-	*HLA-DRB1*, *HLA-DQA1*	
rs113653754-C	1 × 10^−86^	1.3593605	-	*HLA-DQA1*, *HLA-DQB1*	
rs9268877-?	4 × 10^−23^	-	-	*HLA-DRB9*	[91]
rs9271366-G	1 × 10^−18^	2.1	[1.78–2.48]	*HLA-DRB1*, *HLA-DQA1*	[88]
rs9268853-T	1 × 10^−55^	1.4	[1.34–1.47]	*HLA-DRB9*	[67]
rs2395185-G	5 × 10^−22^	1.92	[1.68–2.19]	*HLA-DRB9*	[68]
rs9268923-C	4 × 10^−15^	1.45	[1.33–1.59]	*HLA-DRB9*	[71]
rs2395185-?	1 × 10^−16^	1.52	-	*HLA-DRB9*	[86]
rs6927022-A	5 × 10^−133^	1.444	[1.387–1.503]	*HLA-DQA1*	[75]
rs2395185-G	9 × 10^−23^	1.49	[NR]	*HLA-DRB9*	[83]
rs9268877-T	6 × 10^−18^	1.45	[1.33–1.58]	*HLA-DRB9*	[70]
rs6927022-?	5 × 10^−65^	-	-	*HLA-DQA1*	[69]
rs117506082-G	4 × 10^−88^	3.39	[2.99–3.83]	*HCG27*, *HLA-C*	[100]
rs2239805-C	1 × 10^−10^	0.839	[0.796–0.885]	*HLA-DRA*	[78]
rs2239805-C	7 × 10^−9^	0.895	[0.862–0.929]	*HLA-DRA*	
rs9274238-A	6 × 10^−14^	-	[0.23–0.38]	*HLA-DQB1*	[81]
rs147732109-A	2 × 10^−16^	-	[0.96–1.56]	*HLA-DRB6*, *HLA-DRB1*	[85]
rs9271511-?	1 × 10^−158^	-	[0.32–0.37]	*HLA-DRB1*, *HLA-DQA1*	[80]
HLA-DRB1*0103-?	2 × 10^−13^	6.9418	[6.43–7.46]	*-*	[101]
HLA-DRB1*1301-?	7 × 10^−8^	2.1073	[1.84–2.38]	*-*	

**Table 2 genes-16-01085-t002:** Polymorphisms and their therapy implications [98,118,119,120,121,122].

Variant and Risk Allele (rs)	Gene	Therapy Observations	References
rs10889677	*IL23R*	positive response to infliximab	[118]
rs11209032			
rs1004819-AA			
rs2201841-GC			
rs1495965-CC			
rs1343151-AA	*IL23R*	decreased response to infliximab	
rs7517847-GC			
rs11465804-CC			
rs10489629-CC			
rs5030728-A	*TLR4*	positive response to infliximab	[119]
rs11465996	*LY96*	subtherapeutic infliximab levels	
rs5030728-GG	*TLR4*	subtherapeutic infliximab levels	
rs3397-T	*TNFRSF1B*	non-response to infliximab	
rs2569190-A	*CD14*	lack of response to infliximab and adalimumab	
HLA-DRB10103	*HLA-DRB1*	immunogenicity to infliximab	[98]
HLA-DQA105	*HLA-DQA1*	increased anti-drug antibody formation therapy failure	
	*NUDT15*	azathioprine-induced leukopenia	[120,121,122]

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
