# Peer review of "Genetic and Epigenetic Factors in Ulcerative Colitis: A Narrative Literature Review"

_genes, 2025, doi:10.3390/genes16091085_

Round 1
Reviewer 1 Report
Comments and Suggestions for Authors
Overall comments
The conceptual scientific necessity for a manuscript such as this is welcome and-perhaps-timely. Would the authors please think about the remarks made in the “Major comments” section below from a prospective/ spirit of academic helpfulness-rather than negativity in relation to improving the manuscript. Even though the aforementioned involves major changes to the current manuscript rather than minor changes.
Minor corrections
There are, unfortunately, countless mistakes in grammar/syntax throughout the manuscript (I am aware that the first language of the authors is not likely to be English).
1) Line 20; remove the terms “the” and “one” from “...the genetic one….”
2) Line 21; remove the term “….and….” replace with a semicolon (:)
3) Line 27; Ulcerative Coliti is misspelt
4) Line33; “According with Kaplan and Windsor,…..”; replace the word with by “to”
5) Line 38/39; “In last decades, an increase in the incidence of IBD has been observed
worldwide”; replace with: “In the last three decades, an increase in the incidence of
IBD has been detected worldwide”
6) Line 43; “….to almost a double number of 6,8 milion in 2017.“ Replace the foregoing by”
“….to almost double that number of 6,8 milion in 2017.“
7) Line 44; Replace “The prevalence in USA was …….” by “The prevalence in the USA
was……”
8) Lines 44/45; Replace “Thus, in USA around 1 million peo-44 ple have IBD, while in
Europe, it is estimated that around 2.5 – 3 million people are affected by it [5,8].
Zhang……” with ““Thus, in the USA around 1 million people have IBD, while in Europe,
it is estimated that around 2.5 – 3 million people are affected by the disease[5,8].
Zhang……”
9) Line 48; replace “….USA …..” with: “….the USA …..”
(First 50 lines of the manuscript alone).
…………………………………………………………………………………………………..
Reference 71 is not aligned in Table 2
…………………………………………………………………………………………………..
Major comments
1. Is the title of the manuscript appropriate given that it does not involve any experimental work? Perhaps the title should be changed to:
“Genetic and Epigenetic Factors in Ulcerative Colitis: A Narrative Literature Review”
2. Would it not be helpful to the general reader-who is not a specialist geneticist in the area of UC to associate the data in the Tables with the impact of the genes referred to with, for example, the (i) immune system, (ii) the gut (health/integrity) and (iii) other areas of health/diagnosis/research associated with UC?
3) In section 3.2 of the manuscript, should there not be a detailed discussion of the epigenetic role of histones and other non-coding RNAs (other than those mentioned e.g., lncRNAs ) in UC? The foregoing relates particularly-but not exclusively-to factors such as susceptibility to UC, progress of the disease, biomarkers and possible therapeutic interventions.
4. The authors may wish to consult the following references in order to assess whether or not the inclusion of some of the references and the data therein helps to improve their manuscript?
https://www.nature.com/articles/s41598-025-95125-4
https://www.nature.com/articles/s41598-023-33292-y?fromPaywallRec=false
https://www.nature.com/articles/s41598-024-65481-8?fromPaywallRec=false
5. If possible , it is suggested that the authors include one or two useful/necessary Figures in order to highlight the scientific content of the manuscript? Currently the manuscript is somewhat “flat” in terms of being of interest to readers in the subject matter under consideration..
Comments on the Quality of English Language
The quality of the English (grammar/syntax) in the manuscript definetly needs to be improved.
Author Response
“The conceptual scientific necessity for a manuscript such as this is welcome and-perhaps-timely. Would the authors please think about the remarks made in the “Major comments” section below from a prospective/ spirit of academic helpfulness-rather than negativity in relation to improving the manuscript. Even though the aforementioned involves major changes to the current manuscript rather than minor changes.
Answer: Thank you very much for your comments and constructive observations.
Minor corrections
There are, unfortunately, countless mistakes in grammar/syntax throughout the manuscript (I am aware that the first language of the authors is not likely to be English).
Answer: Indeed, English it is not our first language. We have made multiple changes in the manuscript regarding the grammar.
- Line 20; remove the terms “the” and “one” from “...the genetic one….”
Answer: We have made the requested modifications.
- Line 21; remove the term “….and….” replace with a semicolon (:)
Answer: We have made the requested modifications.
- Line 27; Ulcerative Coliti is misspelt
Answer: We have made the requested modifications.
- Line33; “According with Kaplan and Windsor,…..”; replace the word with by “to”
Answer: We have made the requested modifications.
5) Line 38/39; “In last decades, an increase in the incidence of IBD has been observed worldwide”; replace with: “In the last three decades, an increase in the incidence of IBD has been detected worldwide”
Answer: We have made the requested modifications.
6) Line 43; “….to almost a double number of 6,8 milion in 2017.“ Replace the foregoing by” “….to almost double that number of 6,8 milion in 2017.“
Answer: We have made the requested modifications.
7) Line 44; Replace “The prevalence in USA was …….” by “The prevalence in the USA was……”
Answer: We have made the requested modifications.
8) Lines 44/45; Replace “Thus, in USA around 1 million peo-44 ple have IBD, while in Europe, it is estimated that around 2.5 – 3 million people are affected by it [5,8]. Zhang……” with ““Thus, in the USA around 1 million people have IBD, while in Europe, it is estimated that around 2.5 – 3 million people are affected by the disease[5,8]. Zhang……”
Answer: We have made the requested modifications.
9) Line 48; replace “….USA …..” with: “….the USA …..”
Answer: We have made the requested modifications.
Reference 71 is not aligned in Table 2
Answer: We have aligned the reference and moved the table in supplementary materials.
Major comments
- Is the title of the manuscript appropriate given that it does not involve any experimental work? Perhaps the title should be changed to: “Genetic and Epigenetic Factors in Ulcerative Colitis: A Narrative Literature Review”
Answer: We have changes the manuscrip title with: “Genetic and Epigenetic Factors in Ulcerative Colitis: A Narrative Literature Review”
- Would it not be helpful to the general reader-who is not a specialist geneticist in the area of UC to associate the data in the Tables with the impact of the genes referred to with,for example, the (i) immune system, (ii) the gut (health/integrity) and (iii) other areas of health/diagnosis/research associated with UC?
Answer: We have integrated some details about some specific genes and their impact in ulcerative colitis in the text, and we moved the table in the supplementary materials.
3) In section 3.2 of the manuscript, should there not be a detailed discussion of the epigenetic role of histones and other non-coding RNAs (other than those mentioned e.g., lncRNAs ) in UC? The foregoing relates particularly-but not exclusively-to factors such as susceptibility to UC, progress of the disease, biomarkers and possible therapeutic interventions.
Answer: We have added 2 subsections about de lncRNA and histones modifications in ulcerative colitis.
- The authorsmay wish to consult the following references in order to assess whether or not the inclusion of some of the references and the data therein helps to improve their manuscript?
https://www.nature.com/articles/s41598-025-95125-4
https://www.nature.com/articles/s41598-023-33292-y?fromPaywallRec=false
https://www.nature.com/articles/s41598-024-65481-8?fromPaywallRec=false
Answer: Thank you for your articles suggestions. We have added them in the manuscript.
- If possible , it is suggested that the authors include one or two useful/necessary Figures in order to highlight the scientific content of the manuscript? Currently the manuscript is somewhat “flat” in terms of being of interest to readers in the subject matter under consideration.
Answer: Thank you suggestions. We have added a figure about the pathophysiology in UC, as well as two other tables (more simple than the last ones).
Reviewer 2 Report
Comments and Suggestions for Authors
In this review, Caba L et al. discuss the pathophysiology of ulcerative colitis (UC), and common GWAS loci from GWAS catalog and epigenetic factors, namely DNA methylation changes and miRNAs, associated with UC. Increasing number of studies on inflammatory bowel disease and UC genetics necessitates a comprehensive summary of recent findings, therefore, this review might be beneficial to address this need. The section discussing the importance of genetic variants in personalized therapies is a good contribution. However, the content and structure of the manuscript and the scope described in the abstract and title are not quite consistent. Also, it is not very clear what specific insight or discussion that this review contributes to. I would suggest addressing the following points:
- Although the title indicates that this review is on both genetic and epigenetic factors of UC, there is no information regarding epigenetics in the abstract.
- Introduction contains information on clinical findings, diagnosis criteria and treatment, which are too detailed and, in my opinion, not in the scope of the current review and journal. In contrast, it includes too little information on genetic factors and no information on epigenetics of UC, which are the focus of this manuscript.
- It is not very clear what unique insights/discussions that this review contributes to. It would be good to add a couple of sentences at the end of the introduction section to describe what this review will focus on.
- Section 2 consists of detailed pathophysiology of UC, which is probably not the main point of a review on genetics and epigenetics of UC. I think this needs to be discussed together with reported genetic/epigenetic factors or the title and abstract should include that this is also a review on UC pathophysiology.
- Section 3 includes two long tables that are difficult to read and navigate including gene names harboring significant variants (adding a P value threshold would be helpful) associated with UC and a table listing variants and studies. Although these tables can be useful for rapid review of previously reported genes and variants, readers can just go to GWAS catalog and retrieve this data themselves as well. I think a review on genetic factors contributing to a disease needs to include a discussion on reported genes or variants. Therefore, it would be good to move the tables, especially the Table 2 to the supplementary material, and discuss prominent genes/variants (e.g., based on mechanistic insights and/or previous fine mapping results) together with known mechanistic insights instead of providing separate pathophysiology and genetics sections.
- It is not clear how UC associations were retrieved from the GWAS catalog. Did the authors use only “ulcerative colitis” while searching for the associated variants and genes? There are several shared associations of inflammatory diseases that would indicate pleiotropy and are relevant, such as NOD2 in ankylosing spondylitis, psoriasis, UC, Crohn’s disease and primary sclerosing cholangitis, but as far as I see, such associations were not included in tables.
- Adding cohort information from original studies (sample size and population), especially in the variant table would be useful for the readers while interpreting current knowledge.
- I think a separate subsection discussing HLA region would be needed in Section 3 considering its role in UC.
- Since the title includes genetic factors, it would be a good addition to include a section related to the role of rare genetic variation in UC pathogenesis. There are several published studies that used sequencing datasets to identify such variants. Otherwise, it would be good to revise the title as common genetic variation and epigenetic factors in ulcerative colitis.
- It would be helpful to also add a brief section discussing monogenic IBD genes in very early onset IBD and how they differ from those identified through GWAS of UC patients. Or it can be addressed in a paragraph as a limitation/non-focus of the review.
- Section 4 on importance of genetic variants in personalized therapies is a nice addition, therefore, if possible, expanding that section and generating a table or a figure would be helpful for readers.
Author Response
In this review, Caba L et al. discuss the pathophysiology of ulcerative colitis (UC), and common GWAS loci from GWAS catalog and epigenetic factors, namely DNA methylation changes and miRNAs, associated with UC. Increasing number of studies on inflammatory bowel disease and UC genetics necessitates a comprehensive summary of recent findings, therefore, this review might be beneficial to address this need. The section discussing the importance of genetic variants in personalized therapies is a good contribution. However, the content and structure of the manuscript and the scope described in the abstract and title are not quite consistent. Also, it is not very clear what specific insight or discussion that this review contributes to. I would suggest addressing the following points:
Answer: Thank you very much for your comments and constructive observations.
- Although the title indicates that this review is on both genetic and epigenetic factors of UC, there is no information regarding epigenetics in the abstract.
Answer: We have added a phrase about epigenetics in the abstract.
- Introduction contains information on clinical findings, diagnosis criteria and treatment, which are too detailed and, in my opinion, not in the scope of the current review and journal. In contrast, it includes too little information on genetic factors and no information on epigenetics of UC, which are the focus of this manuscript.
Answer: We have deleted some informations about the diagnostic (which indeed was a little too detailed) and we adeed some nome informations regarding genetics and epigenetics in ulcerative colitis.
- It is not very clear what unique insights/discussions that this review contributes to. It would be good to add a couple of sentences at the end of the introduction section to describe what this review will focus on.
Answer: We introduced a sentence with the aim of our review.
- Section 2 consists of detailed pathophysiology of UC, which is probably not the main point of a review on genetics and epigenetics of UC. I think this needs to be discussed together with reported genetic/epigenetic factors or the title and abstract should include that this is also a review on UC pathophysiology. – nu renuntam, explicam de ce nu renuntam la discutii
Answer: We chose to discuss in detail the pathophysiology of ulcerative colitis, since this is essential for understanding the role and the mechanism through which genes lead to the development of ulcerative colitis.
- Section 3 includes two long tables that are difficult to read and navigate including gene names harboring significant variants (adding a P value threshold would be helpful) associated with UC and a table listing variants and studies. Although these tables can be useful for rapid review of previously reported genes and variants, readers can just go to GWAS catalog and retrieve this data themselves as well.
I think a review on genetic factors contributing to a disease needs to include a discussion on reported genes or variants. Therefore, it would be good to move the tables, especially the Table 2 to the supplementary material, and discuss prominent genes/variants (e.g., based on mechanistic insights and/or previous fine mapping results) together with known mechanistic insights instead of providing separate pathophysiology and genetics sections.
Answer: We have added some details about some of the best-known genes and the mechanisms through which they exert their effects. We have also moved both Table 1 and Table 2 to the supplementary materials.
- It is not clear how UC associations were retrieved from the GWAS catalog. Did the authors use only “ulcerative colitis” while searching for the associated variants and genes? There are several shared associations of inflammatory diseases that would indicate pleiotropy and are relevant, such as NOD2 in ankylosing spondylitis, psoriasis, UC, Crohn’s disease and primary sclerosing cholangitis, but as far as I see, such associations were not included in tables.
Answer: Yes, indeed the search was only for the genes related strictly to ulcerative colitis. We tried to exclude the association with other diseases, especially with Crohn disease.
- Adding cohort information from original studies (sample size and population), especially in the variant table would be useful for the readers while interpreting current knowledge.
Answer: We have added the cohort informations from the originalstudies (in the footnote of the Table 2, which was moved into supplementary materials).
- I think a separate subsection discussing HLA region would be needed in Section 3 considering its role in UC.
Answer: We have added a separate short section about HLA region, as well as a table regarding the significant variants in this region.
- Since the title includes genetic factors, it would be a good addition to include a section related to the role of rare genetic variation in UC pathogenesis. There are several published studies that used sequencing datasets to identify such variants. Otherwise, it would be good to revise the title as common genetic variation and epigenetic factors in ulcerative colitis.
Answer: We added a short section about some rare variants in UC, although we found more informations about rare variants in IBD (generalized) or with focus on Crohn disease, which were not exactly the focus of this article.
- It would be helpful to also add a brief section discussing monogenic IBD genes in very early onset IBD and how they differ from those identified through GWAS of UC patients. Or it can be addressed in a paragraph as a limitation/non-focus of the review.
Answer: We have added a short section about Monogenic UC.
- Section 4 on importance of genetic variants in personalized therapies is a nice addition, therefore, if possible, expanding that section and generating a table or a figure would be helpful for readers.
Answer: We added more details about some variants implications in treatment, as well as some details about HLA variants and lncRNA implications in therapy. We also generated a table which include these aspects.
Round 2
Reviewer 1 Report
Comments and Suggestions for Authors
I thank the authors for taking into account my comments relating to the first draft of the manuscript in terms of the modifications made to V2 of the manuscript.
Overall V2 of the manuscript is-bar any comments/views of the editors of the journal-now
"fit for purpose".
Reviewer 2 Report
Comments and Suggestions for Authors
The revised manuscript has been significantly improved to address the points that were previously missing, and I believe it is now ready for publication.